# Plasma-Derived Exosome Proteins as Novel Diagnostic and Prognostic Biomarkers in Neuroblastoma Patients

**DOI:** 10.3390/cells12212516

**Published:** 2023-10-25

**Authors:** Martina Morini, Federica Raggi, Martina Bartolucci, Andrea Petretto, Martina Ardito, Chiara Rossi, Daniela Segalerba, Alberto Garaventa, Alessandra Eva, Davide Cangelosi, Maria Carla Bosco

**Affiliations:** 1Laboratory of Experimental Therapies in Oncology, IRCCS Istituto Giannina Gaslini, 16147 Genova, Italy; martinamorini@gaslini.org (M.M.); martinaardito@gaslini.org (M.A.); danielasegalerba@gaslini.org (D.S.); 2Unit of Autoinflammatory Diseases and Immunodeficiencies, Pediatric Rheumatology Clinic, IRCCS Istituto Giannina Gaslini, 16147 Genova, Italy; chiararossi@gaslini.org (C.R.); mariacarlabosco@gaslini.org (M.C.B.); 3Core Facilities, Clinical Proteomics and Metabolomics, IRCCS Istituto Giannina Gaslini, 16147 Genova, Italy; martinabartolucci@gaslini.org (M.B.); andreapetretto@gaslini.org (A.P.); 4Pediatric Oncology, IRCCS Istituto Giannina Gaslini, 16147 Genova, Italy; albertogaraventa@gaslini.org; 5Scientific Directorate, IRCCS Istituto Giannina Gaslini, 16147 Genova, Italy; alessandraeva@gaslini.org; 6Clinical Bioinfomatics Unit, IRCCS Istituto Giannina Gaslini, 16147 Genova, Italy; davidecangelosi@gaslini.org

**Keywords:** neuroblastoma, exosomes, biomarkers

## Abstract

Neuroblastoma (NB) is the most common extracranial solid tumor during infancy, causing up to 10% of mortality in children; thus, identifying novel early and accurate diagnostic and prognostic biomarkers is mandatory. NB-derived exosomes carry proteins (Exo-prots) reflecting the status of the tumor cell of origin. The purpose of this study was to characterize, for the first time, the Exo-prots specifically expressed in NB patients associated with tumor phenotype and disease stage. We isolated exosomes from plasma specimens of 24 HR-NB patients and 24 low-risk (LR-NB) patients at diagnosis and of 24 age-matched healthy controls (CTRL). Exo-prot expression was measured by liquid chromatography–mass spectrometry. The data are available via ProteomeXchange (PXD042422). The NB patients had a different Exo-prot expression profile compared to the CTRL. The deregulated Exo-prots in the NB specimens acted mainly in the tumor-associated pathways. The HR-NB patients showed a different Exo-prot expression profile compared to the LR-NB patients, with the modulation of proteins involved in cell migration, proliferation and metastasis. NCAM, NCL, LUM and VASP demonstrated a diagnostic value in discriminating the NB patients from the CTRL; meanwhile, MYH9, FN1, CALR, AKAP12 and LTBP1 were able to differentiate between the HR-NB and LR-NB patients with high accuracy. Therefore, Exo-prots contribute to NB tumor development and to the aggressive metastatic NB phenotype.

## 1. Introduction

Neuroblastoma (NB) is the most common extracranial solid tumor during infancy, with a median age at diagnosis of 17 months. Although classified as a rare disease, it represents a worldwide emergency, causing up to 10% of mortality in children. NB shows notable heterogeneity with regard to histology and clinical behavior, ranging from low-risk (LR-NB) localized tumors to high-risk (HR-NB) disease, characterized by an aggressive metastatic phenotype, resistance to treatment, and fatal relapse occurrence. Risk stratification demands the highest accuracy, as it determines the therapeutic treatment. For NB patients, risk assessment is based on age at diagnosis, stage of the disease, chromosomal aberrations and amplification of *MYCN*, a transcriptional factor considered the major oncogenic driver. However, the current therapeutic stratification, based on such clinical and molecular risk factors, does not allow to discriminate patients with similar clinical-pathological parameters who receive the same treatment, despite showing markedly different clinical courses. The limits of the existing classification system could be overcome by a deeper investigation of the biology of the tumor. Thus, the challenge remains the identification of additional and more accurate prognostic markers to improve risk stratification and direct a personalized treatment.

Despite aggressive therapy, the five-year survival rate of HR-NB patients remains poor. Moreover, HR-NB tumors are often unresectable as they infiltrate surrounding areas, hindering tissue biopsies. Nowadays, the molecular characterization of neoplastic diseases allows to overcome the limitations of the cell morphology-based classification of solid tumors, improving cancer diagnosis and treatment [1]. The acquired experience in molecular profiling highlights some limitations, especially in terms of sampling methods [2]. Although they are still the main source of biological material for diagnostic purposes, tissue biopsies entail high costs, high risk of adverse effects because of the invasive nature of the technique and procedural complications resulting in inadequate material retrieval and lead to increased false-negative or false-positive results that affect treatment decisions [3,4]. Moreover, tissue sampling does not capture all of the spatial and temporal heterogeneity of tumors, preventing the identification of aggressive and therapy-resistant cellular subclones that are responsible for disease progression and recurrence [5].

Recently, interest has grown in liquid biopsies, which represent a surrogate of the tumor and can provide complementary or even additional information for tissue biopsies. The investigation of biological fluids is based on a minimally invasive, less expensive and easily accessible method that can be applied for cancer diagnosis and prognosis, treatment selection, disease monitoring and assessment of relapse occurrence. Liquid biopsies are a source of circulating tumor cells, circulating tumor DNA, exosomes, and nano-sized vesicles (30–150 nm) derived from the endocytic pathway and released in the extracellular environment after the fusion of multivesicular bodies with the cell plasma membrane [6]. The release of exosomes is regulated by specific stimuli, and their cargo reflects the producing cell [7]. Indeed, exosomes are defined as the bioprint of the cell of origin and, thus, the study of their content allows to gain accurate information of primary tumor features [8,9,10]. Indeed, exosomes represent a source for both diagnostic and prognostic markers in several human diseases [8,9,10,11,12,13,14,15]. To gain insight into the characteristics of NB for improving diagnosis and treatment, we performed proteomic analysis of the exosomes released in the blood of NB patients. In particular, in order to identify molecules with potential value for improving risk stratification, we compared the exosomal protein (Exo-prot) profiles of LR-NB and HR-NB patients at the onset. Our results identified the Exo-prots and biological pathways involved in tumor development and progression and associated with the acquisition of aggressive tumoral traits, integrating the data of our previous study demonstrating that exosomal microRNAs (exo-miRNAs) are associated with HR-NB patients’ response to treatment [8].

## 2. Materials and Methods

### 2.1. Study Cohort and Blood Sample Collection and Processing

The study cohort included Italian NB patients classified into different risk groups: HR-NB (*n* = 24) and LR-NB (*n* = 24). Age-matched blood samples of healthy donors (*n* = 24) were used as the control (CTRL). The main clinical features of the study cohort are reported in Table 1. The procedures for patient enrollment were carried out according to the proper guidelines and in adherence with the ethical principles of the Declaration of Helsinki. Blood samples were provided by the BIT-Gaslini Biobank, which centralizes the NB specimens derived from Italian AIEOP (Associazione Italiana Emato-Oncologia Pediatrica) centers. Written informed consent was provided from the parents or legal guardian of each patient considered for the study. Blood samples were collected in EDTA tubes and processed within 24 h. Processing included centrifugation at 1200× *g* for 10 min at room temperature (RT) to collect plasma. Plasma was stored at −80 °C or used immediately for exosome isolation.

### 2.2. Exosome Isolation and Proteomic Analysis

Exosome isolation was performed from 500 μL of plasma utilizing an exoRNeasy Serum/Plasma Midi Kit (Qiagen Italia, Milan, Italy) with a modified version of the protocol that we have previously described [8], which enables to isolate vesicles with diameter and surface markers specific to exosomes. Protocol modifications were added starting from the washing step with XWP buffer, performed twice by centrifuging samples at 500× *g* and 5000× *g* for 5 min at RT, respectively. The isolated exosomes were lysed, reduced and alkylated directly on the filter by adding to 100 µL of boiling lysis buffer 6 M Guanidine, 10 mM TCEP, and 4 mM CAA in 100 mM Tris pH 8 and heating the columns in a thermomixer at 100 °C for 10 min and 1000 rpm. The lysed samples were centrifuged at 4000× *g* for 5 min and left to digest for 2 h with the addition of 0.6 µg of LysC (FUJIFILM Wako Pure Chemical Corporation, Milano, Italy). The filters were then washed with 200 μL of Tris 25 mM, pH 8, by centrifuging at 4000× *g* for 5 min. The total sample was further digested by adding 0.6 µg of Trypsin (Promega, Milano, Italy), followed by mixing and incubating at 37 °C overnight. The digested samples were loaded onto StageTips [16]. The resulting peptides were completely dried in a SpeedVac centrifuge at 30 °C, resuspended in 2% ACN and 0.1% formic acid and analyzed by a nano-UHPLC-MS/MS system using an Ultimate 3000 RSLC coupled to an Orbitrap Fusion Tribrid mass spectrometer (Thermo Scientific, Milano, Italy).

### 2.3. Proteomic Set-Up

Elution was performed with an EASY spray column (75 μm × 50 cm, 2 μm particle size, Thermo Scientific, Milano, Italy) at a flow rate of 250 nL/min with a 100 min non-linear gradient of 7–45% solution B (80% ACN, 20% H2O, 5% DMSO and 0.1% FA). MS analysis was performed in DDA mode. Orbitrap detection was used for MS1 measurements at a resolving power of 120 K in a range between 375 and 1500 *m*/*z*, with an AGC target of 400,000 and a maximum injection time of 50 ms. MS/MS spectra were acquired in the linear ion trap (rapid scan mode) after collision-induced dissociation (CID) fragmentation at a collision energy of 35% and an AGC target of 2000 for up to 300 ms. A 1.5 s cycle time was performed for data-dependent MS/MS analysis, during which precursors with a charge range of 2–4 were selected for activation in order of abundance. Quadrupole isolation with a 1.8 *m*/*z* isolation window was used, and dynamic exclusion was enabled for 25 s.

Raw data were processed with MaxQuant [17] software, version 1.6.4.0. A false discovery rate (FDR) of 0.01 was set for the identification of proteins, peptides and PSM (peptide-spectrum match). For peptide identification, a minimum length of 6 amino acids was required. The Andromeda engine, incorporated into MaxQuant software, was used to search the MS/MS spectra against the Uniprot human database and additional human database (release UP000005640_9606 April 2018). In the processing, acetylation (Protein N-Term), oxidation (M) and deamidation (NQ) were selected as variable modifications, and the fixed modification was carbamidomethylation (C). The quantification intensities were calculated by the default fast MaxLFQ algorithm with the activated option “match between runs.”

### 2.4. Bioinformatic Analysis

Differential expression analysis was carried out using the SMAGEXP tool suite [18]. Exo-prots with a Benjamini–Hochberg *p*-value lower than 0.05 and a log fold change >0.58 or <−0.58 were considered statistically significantly deregulated. Exo-prots with ≤30% missing values were retained for the analysis to reduce the bias introduced by imputation. The random forest (RF) and quantile regression imputation of left-censored data (QRILC) methods were used for missing value imputation using the METimp web tool [19]. The RF method uses an iterative imputation approach that trains an RF on observed values and then predicts the missing values [20]. The QRILC method performs the imputation of left-censored missing data, using random draws from a truncated distribution with parameters estimated by quantile regression [19]. Protein–protein interaction (PPI) network and pathway analyses were carried out on selected statistically significantly deregulated Exo-prots using STRING-DB software (version 12.0) [21]. The PPI enrichment *p*-value was used to assess whether a group of proteins have more interactions among themselves than what would be expected from a random set of proteins of a similar size, drawn from the genome. We considered significant a PPI enrichment *p*-value lower than 0.05. Pathway analysis was used to identify the biological processes mainly regulated by differentially expressed Exo-prots. An FDR lower than 0.05 identified significantly enriched ontology terms and pathways. Venn diagrams were plotted using the InteractiVenn web-based tool [22].

### 2.5. Statistical Analysis

A two-proportion Z-test based on the detectable and missing values between the NB and CTRL or HR-NB and LR-NB groups was used to determine whether the two proportions were different from each other. A Z-score <−2.6 or >2.6 were considered to demonstrate statistically significant differences. Age group and sex were considered potential confounders, and differences between the NB patients and CTRL were tested using Fisher’s exact test. A Fisher *p*-value lower than 0.05 was considered statistically significant. Receiver operating characteristic (ROC) curves were plotted to visually display the discriminating power of each significantly modulated Exo-prot across groups. Exo-prot expression was used to generate ROC curves. The area under the ROC curve (AUC) was computed to quantitatively assess the discriminating performance of each Exo-prot. ROC curves and statistical calculations were performed using GraphPad Prism 8.0 software (GraphPad Software, La Jolla, CA, USA).

The combined discriminatory power of a selection of statistically significantly represented Exo-prots across the NB and CTRL groups was assessed by generalized linear models using the GLMNET R package [23]. The group was set up as the response variable, Exo-prot expression as the explanatory variable, the elastic net mixing parameter at 1.0, and the minimum mean square error as the gamma value. The best gamma value was assessed using the leave-one-out cross-validation technique.

### 2.6. ROC Data Validation by Gene Expression

Diagnostic or prognostic Exo-prots identified by ROC analysis were validated by measuring the expression of the corresponding genes in the primary tumor tissue at the onset. Tissue biopsies from 10 LR and 10 HR-NB patients and four non-oncologic control tissues were homogenized with the Tissue Lyser II instrument and RNA was extracted with an RNeasy Mini Kit (Qiagen Italia, Milan, Italy). The RNA quality was assessed with a Nano RNA Assay on the Agilent 2100 Bioanalyzer (Agilent Technologies Spa, Milan, Italy). RNA quantification was performed with the QIAxpert System (Qiagen Italia, Milan, Italy). RNA was reverse-transcribed with SuperScript™ II Reverse Transcriptase (Thermo Fisher Scientific, Milan, Italy) and the expression levels of the *FN1*, *AKAP12*, *MYH9*, *CALR*, *NCL*, *NCAM*, *VASP*, *LGALS3BP*, *DCN* and *LUM* genes were assessed by RTqPCR (real-time quantitative PCR) with specific TaqMan assays (Thermo Fisher Scientific, Milano, Italy). Each sample was repeated in triplicate on a MicroAmp Fast Optical 96-well reaction plate run on the ViiA7 Real-Time PCR System (Thermo Fisher Scientific, Milan, Italy). *GAPDH* was used as the housekeeping gene for data normalization. The relative expression values between sample groups were calculated using the 2^(−DCT)^ method. Statistical analysis and plots were carried out using GraphPad Prism 8.0 for Windows.

## 3. Results

### 3.1. Protein Cargo Profiling of the NB and CTRL Subjects

To identify the potential biomarkers and key biological processes associated with tumor development in NB patients, we performed proteomic analysis of the exosomes isolated from the plasma samples of the NB patients and age-matched CTRL provided by the BIT-Gaslini biobank. Table 1 summarizes the clinical and molecular characteristics of the patients and CTRL. A flowchart summarizing the main steps of the sample collection and bioinformatic analyses is shown in Figure 1. Specifically, plasma samples were collected from 24 LR-NB patients (stage 1, 2, 4 s and *MYCN* non-amplified tumors), 24 HR-NB patients (age >18 months and stage 4 tumor or *MYCN* amplified) and 24 CTRL children. The association between sex (male vs. female) or age group (<18 months vs. >18 months) and the subject group (CTRL) or NB-HR vs. LR-NB was analyzed to assess the potential confounding effects of these characteristics. Fisher’s exact test did not show a significant association between sex or age group and the subject group (*p* > 0.05), thereby excluding the possibility that sex and age group might be confounding factors in this study.

Plasma-derived exosomes from each sample were isolated, and protein cargo was profiled by high-resolution mass spectrometry coupled with liquid chromatography. The analytical data generation identified 458 exosomal proteins (Exo-prots) detectable in at least one NB or CTRL subject.

### 3.2. Analysis of Detectable Exo-Prots Based on the Distribution of Missing Values in the NB and CTRL Subjects

Since missing values can adversely affect statistical analysis [24], the distribution of the Exo-prot missing values between the NB patients and CTRL was investigated to assess whether detectable Exo-prots might provide new insights into disease development. The Z-test analysis of the detectable value proportions showed that 38 Exo-prots were exclusively expressed in the NB patients, whereas 83 were exclusively expressed in the CTRL subjects (z-score >2.6 or <−2.6; Appendix A). A heatmap based on the expression of these 121 Exo-prots visually shows a clear association with the NB patients and CTRL subjects, visually confirming the results of the Z-test analysis of proportions (Figure 2A). Among the exclusively detectable Exo-prots in the NB patients, we identified the MYC target nucleophosmin (NPM1) and the actin interacting protein zyxin (ZYX), known for their role in tumor proliferation and invasiveness [25,26].

Network analysis was carried out on the 121 Exo-prots to help interpret the results and assess the biological function of these proteins (Figure 2B). The network analysis displayed significantly more interactions than expected for a random set of proteins of a similar size drawn from the genome (protein–protein interaction (PPI) enrichment *p*-value <0.05). These findings establish a connection between significantly modulated Exo-prots and indicate that these proteins are biologically connected as a group. Pathway analysis is a well-known bioinformatic tool that is used to explore the biological processes and pathways associated with a list of differentially expressed genes/proteins using curated ontologies [27].

Pathway analysis identified 141 significantly enriched biological processes and pathways in the NB patients (Appendix A). A selection of the most significantly enriched terms are listed in Table 2. These results show that the identified Exo-prots in the NB patients might act as cancer-associated pathways, such as stress response, immune system regulation, inflammation, cell motility and cytoskeletal rearrangements.

Missing values cannot be directly handled by bioinformatics tools. Imputation is a bioinformatic technique used to substitute missing values with defined observations. We applied RF or QRILC, two well-known methods [19], for imputing missing values to subsequently identify deregulated Exo-prots.

### 3.3. Differentially Expressed Exo-Prots in the NB Patients

Exo-prots showing at most 30% missing values were retained for imputation, while the rest were filtered out (See Section 2.4). The two imputation methods successfully substituted missing values with numeric values, thus defining two distinct datasets, hereafter referred to as the RF and QRILC datasets. Principal component analysis based on the protein expression profile of 24 HR-NB, 24 LR-NB and 24 CTRL subjects was performed in order to assess the presence of outlier samples. The results indicate that subject CTRL_18 was a potential outlier exclusively in the QRLIC dataset, but was not in the RF dataset. Therefore, we concluded that QRLIC imputation could have had a substantial impact on the proteomic profile of this specific subject. In order to identify specific deregulated Exo-prot biomarkers in the NB patients, we compared the Exo-prot expression profile of the NB and CTRL subjects using the SMAEXP tool in the RF and QRILC datasets. In the RF dataset, differential expression analysis identified 38 upregulated and 90 downregulated Exo-prots in the NB patients compared to the CTRL subjects (Appendix A), whereas in the QRILC dataset, 32 upregulated and 92 downregulated Exo-prots were identified (Appendix A). A Venn diagram of the statistically significantly modulated Exo-prots in the two datasets showed that 17 upregulated and 64 downregulated Exo-prots in the NB patients were shared by the two datasets. Meanwhile, 21 upregulated and 26 downregulated Exo-prots were exclusive to the RF dataset and 15 upregulated and 28 downregulated Exo-prots were exclusive to the QRILC dataset, respectively (Figure 3A). A complete list of the 17 upregulated and 64 downregulated Exo-prots in the NB patients identified in both datasets is reported in Table 3. Heatmap visualization based on the expression values of the 81 Exo-prots commonly regulated in the RF and QRILC datasets shows a clear separation between the NB and CTRL groups (Figure 3B,C). Notably, among the upregulated Exo-prots, we identified proteins that positively regulate tumor development and progression, such as neural cell adhesion molecule (NCAM) [28], nucleolin (NCL) [29] and galectin-3-binding protein (LGALS3BP) [30]. Among the downregulated Exo-prots, we identified proteins with oncosuppressive functions, such as lumican (LUM) [31], decorin (DCN) [32] and the vasodilator stimulated phosphoprotein (VASP) [33]. Network analysis was carried out on the 17 upregulated and 64 downregulated Exo-prots (Figure 3D), showing their strong interactions (PPI enrichment *p*-value <0.05). These findings establish various interactions between significantly modulated Exo-prots in Table 2 and indicate that these proteins are biologically connected as a group. Pathway analysis identified 153 significantly enriched biological processes and pathways in the NB patients (Appendix A). A selection of the most significantly enriched terms are listed in Table 4. These results show that the NB patients were characterized by aberrant expression of proteins involved mainly in inflammation and complement activation, immune response, cell proliferation and apoptosis and extracellular matrix (ECM) interactions.

### 3.4. Analysis of the Detectable Exo-Prots in HR-NB or LR-NB Based on the Distribution of Missing Values

To identify detectable Exo-prots in the HR-NB patients, we compared the Exo-prot expression profiles of the HR-NB and LR-NB patients. A Z-test was used to identify Exo-prots whose distribution of missing values was significantly different between the HR-NB and LR-NB patients. This analysis showed that 63 Exo-prots were exclusively expressed in the HR-NB patients, whereas 52 were exclusively expressed in the LR-NB patients (Z-score >2.6 or <−2.6; Appendix A). A heatmap based on the expression of these 115 Exo-prots showed a clear discrimination between the outcome groups, confirming the results of the Z-test analysis of proportions (Figure 2C). Among the exclusively detectable Exo-prots, we identified well-known inflammation biomarkers such as CPR, SAA1/SAA2 and PTX3 [34,35]. Network analysis showed statistically significant interactions among 115 Exo-prots (PPI enrichment *p*-value <0.05; Figure 2D). Pathway analysis identified 205 significantly enriched processes in the NB patients (Appendix A). A selection of the most significantly enriched terms are listed in Table 5. These results show that the identified Exo-prots in the HR-NB patients are mainly involved in immune responses, inflammation, cell migration, apoptosis and cytoskeletal organization.

### 3.5. Differentially Expressed Exo-Prots in the HR-NB Patients

To identify deregulated Exo-prots in the HR-NB patients, we analyzed the Exo-prot expression profiles between the HR-NB and LR-NB patients using the SMAEXP tool in the RF or QRILC dataset. In the RF dataset, the analysis identified 98 upregulated and 70 downregulated Exo-prots in the HR-NB patients compared to the LR-NB patients (Appendix A), whereas in the QRILC dataset, 58 upregulated and 59 downregulated Exo-prots were identified (Appendix A). A Venn diagram of the deregulated Exo-prots in the two datasets showed that 46 upregulated and 46 downregulated Exo-prots in the HR-NB patients were common between the two datasets. On the contrary, 52 upregulated and 24 downregulated Exo-prots were exclusive to the RF dataset and 12 upregulated and 13 downregulated Exo-prots were exclusive to the QRILC dataset, respectively (Figure 4A). A complete list of 46 upregulated and 46 downregulated Exo-prots in the HR-NB patients identified in both datasets is reported in Table 6. Heatmap visualization of the expression values shows a clear separation between the HR-NB and LR-NB groups (Figure 4B,C). In the HR-NB patients, we observed the upregulation of proteins that positively regulate cell migration and metastasis: Myosin-9 (MYH9) [36,37], fibronectin (FN1) [38] and latent-transforming growth factor-beta-binding protein 1 (LTBP1) [39]. Among the downregulated Exo-prots, we identified proteins exerting antitumoral effects: Calreticulin (CALR) [40] and A-kinase anchor protein 12 (AKAP12) [41,42,43]. Network analysis was carried out on the 46 upregulated and 46 downregulated Exo-prots in the HR-NB patients (Figure 4D). These findings establish statistically significant interactions among the deregulated Exo-prots in Table 6 and indicate that these proteins are biologically connected as a group (PPI enrichment *p*-value <0.05). Pathway analysis identified 482 significantly enriched biological processes and pathways in the HR-NB patients (Appendix A). A selection of the most significantly enriched terms is reported in Table 7. These results show that the Exo-prots modulated in the HR-NB patients are involved not only in inflammation, immune response and cell proliferation, but also specifically in cell migration, adhesion and motility, as well as in angiogenic processes. Our data suggest that Exo-prots may promote the metastatic properties of HR-NB tumors.

### 3.6. ROC Analysis of the Diagnostic Value of the Exo-Prots in the NB and CTRL Subjects and the Prognostic Significance of the Exo-Prots in the HR-NB and LR-NB Patients

In order to assess the sensitivity and specificity of the potential biomarkers, we performed ROC analysis on the selected Exo-prots among those found to be significantly differentially expressed in the NB and HR-NB patients. Analysis was performed using both the RF and QRILC datasets. NCAM, NCL, LGALS3BP, LUM, DCN and VASP were studied for their potential diagnostic value in NB, whereas MYH9, FN1, LTBP1, CALR and AKAP12 were investigated for their prognostic value in HR-NB. The area under the curve (AUC) was used to measure the degree of separability of the Exo-prots between the NB and CTRL or HR-NB and LR-NB patients. The results show that NCAM (RF: AUC 0.83; QRILC: AUC 0.83, both *p* < 0.0001; Figure 5A), NCL (RF: AUC 0.85, QRILC: AUC 0.83, both *p* < 0.0001; Figure 5B), LUM (RF: AUC 0.77, *p* = 0.0002/QRILC: AUC 0.77, *p* = 0.0002; Figure 5C) and VASP (RF: AUC 0.8, QRILC: AUC 0.85, both *p* < 0.0001; Figure 5D) had the highest AUCs and *p*-values, indicating a robust and significant diagnostic value for NB patients. Meanwhile, LGALS3BP (RF: AUC 0.65, *p* = 0.03/QRILC: AUC 0.65, *p* = 0.03; Figure 5E) and DCN (RF: AUC 0.75, *p* = 0.0005/QRILC AUC 0.71, *p* = 0.003; Figure 5F) have a low but significant AUC, indicating a lower diagnostic value. We also investigated whether the combination of Exo-prots could improve the discriminatory power of these markers taken singularly. We applied the generalized linear model (GLMNET) algorithm (see Material and Methods). Our findings show that the discriminatory power of the combined model was higher than that obtained by each biomarker taken singularly in both datasets (RF: AUC 0.97, QRILC: AUC 0.97, both *p* < 0.0001; Figure 5G).

Comparing the HR-NB and LR-NB samples, ROC analysis evidenced that MYH9 (RF: AUC 0.95, QRILC: AUC 0.92, both *p* < 0.0001; Figure 6A), FN1 (RF: AUC 0.89, QRILC: AUC 0.89, both *p* < 0.0001; Figure 6B) and CALR (RF: AUC 0.94, QRILC: AUC 0.94, both *p* < 0.0001; Figure 6C) had high AUCs and low *p*-values, indicating a robust and significant prognostic value. LTBP1 (RF: AUC 0.78, *p* = 0.0007/QRILC AUC 0.71, *p* = 0.01; Figure 6D) and AKAP12 (RF: AUC 0.74; *p* = 0.005/QRILC: AUC 0.84, *p* < 0.0001; Figure 6E) had lower AUCs, but were still significant, indicating a lower prognostic value. Similarly, as performed between the NB and CTRL groups, we assessed whether the combination of Exo-prots could improve the discriminatory power of these markers taken singularly. We applied the GLMNET algorithm and showed that, again, the discriminatory power of the combined model was higher than that obtained by each single marker in both datasets (RF: AUC 0.99, QRILC: AUC 0.97, both *p* < 0.0001; Figure 6F).

In conclusion, these results indicate that the expression of exosomal NCAM, NCL, LUM and VASP has evident diagnostic value in discriminating NB patients from CTRL subjects, both taken singularly or in combination, suggesting that these non-invasive Exo-prot biomarkers may be used as novel diagnostic tools and provide new insights in the mechanisms of neuroblastoma pathogenesis. Meanwhile, the expression of MYH9, FN1, LTBP1, CALR and AKAP12 is effective in discriminating HR-NB from LR-NB patients, indicating that these non-invasive biomarkers may be used to detect patients at risk of developing aggressive tumor phenotypes.

### 3.7. Validation of the Key Selected Proteins by Gene Expression

To validate the diagnostic and prognostic Exo-prots identified by ROC analysis, we measured the expression of the corresponding coding genes in tissue biopsies of the NB primary tumors of 10 HR and 10 LR cases. We used tissue samples of age-matched non-oncologic subjects as the control. *NCL* and *NCAM* were confirmed as diagnostic markers, significantly overexpressed in NB tumors compared to control samples (*p* < 0.01 and *p* < 0.05, respectively). The diagnostic power of *DCN* and *LUM* was confirmed as well, with a significant downregulation of both markers in NB patients (*p* < 0.001 and *p* < 0.01, respectively) (Figure 7A). *VASP* did not show any significant modulation between the two biological groups considered, while *LGALS3BP*, contrary to the exosome data, was downregulated in the NB primary tumor tissues (Figure 7A). From a prognostic point of view, significant *FN1* overexpression (*p* < 0.05) was confirmed in the HR-NB compared to LR-NB patients, while *MYH9* and *LTBP1* did not report any significant modulation (Figure 7B). In agreement with the proteomic data, *AKAP12* and *CALR* were significantly decreased in HR disease (*p* < 0.05 and *p* < 0.01, respectively) (Figure 7B).

## 4. Discussion

In the present study, we characterized the protein cargo of the exosomes derived from patients with LR-NB and HR-NB tumors and age-matched control subjects to identify novel non-invasive diagnostic or prognostic biomarkers.

We applied LC-MS-based proteomics, a technique that has increased the knowledge about the protein content of extracellular vesicles in recent years [44]. In vitro proteomics studies on neuroblastoma-derived exosomes have been conducted to determine their role in tumor progression and metastasis formation [45,46,47,48]. However, these studies have not been carried out on patient-derived samples, thus limiting their translational applicability to the clinical settings.

The proteomic profile of each patient and CTRL subject was analyzed with two distinct bioinformatic approaches—the distribution of missing values and differential expression analysis. Distribution analysis is expected to detect Exo-prots in which missing values are exclusively detectable in one of the biological groups, whereas differential expression analysis aims to assess the deregulation of Exo-prot expression in a large majority of samples. We considered that missing value distribution analysis could provide useful insights into tumor development because Exo-prots that are undetectable or detectable only in NB patients could actually have diagnostic value and potentially a biological meaning. Distribution analysis can identify distinct Exo-prots from those found via differential expression analysis because no filter is applied to the number of missing values. On the contrary, the differential expression analysis was based on Exo-prots with a maximum of 30% missing data that underwent missing value imputation. Thus, the results of both analyses were complementary.

Analysis of the distribution of missing values was instrumental to identify Exo-prots in NB patients involved in the stress response, immune system regulation, inflammation, cell motility and cytoskeletal rearrangements. Among the exclusively detectable Exo-prots in the NB patients, we identified the nucleolar protein NPM1 as being overexpressed in hematologic and solid tumors [49] and ZYX involved in focal adhesion and in the regulation of actin cytoskeleton with a role in cell motility and tumor invasion [50].

The comparison between HR-NB and LR-NB patients identified the C-reactive protein (CRP), recognized as a biomarker of inflammation, whose elevated serum level has been shown to be associated with poor prognosis in many cancers and in NB [51]; serum amyloid A1/2 (SAA1/SAA2), which have been found to be prognostic in many solid tumors, and whose expression, in combination with high level of CRP, can stratify patients with high-risk early-stage melanoma [34]; pentraxin 3 (PTX3), which is reported to be associated with the PI3K/AKT/mTOR signaling pathway to induce tumor cell proliferation and apoptosis and whose overexpression results in poor survival outcomes in many tumors [52]. Although the missing value distribution analysis suggested the pro-tumoral activity of deregulated Exo-prots, missing data can negatively affect statistical power [24]. Thus, imputation of the missing values is usually performed to obtain robust results [24]. Since RF and QRILC are indicated among the most accurate imputation methods [19], we decided to use both approaches to perform missing value imputation.

Differential expression analysis was instrumental in identifying Exo-prots in the NB patients involved in tumor-promoting processes, such as inflammation and complement activation, immune response, cell proliferation and apoptosis and ECM interaction. Among the upregulated Exo-prots, we identified NCAM, an adhesion molecule known to be highly expressed in NB tumors and associated with increased metastatic capacity [28], and LGALS3BP, a glycosylated protein that has migratory-promoting activity [30]. Moreover, NCL was upregulated in the NB patients; this phosphoprotein regulates several biological processes, including cell proliferation and apoptosis, DNA and RNA metabolism, ribosomal RNA maturation and ribosome assembly [29]. Of note, a high expression of NCL has recently been demonstrated to be an unfavorable independent prognostic factor in NB [53]. Hence, the demonstration of NCL overexpression in exosomes released in the circulation of patients might be relevant to better understand its role in driving NB tumor progression. Among the downregulated proteins, we identified two proteoglycans belonging to the small leucine-rich proteoglycan (SLRP) family: LUM and DCN. The former exerts an oncosuppressive function by impairing metalloproteinase 14 (MMP-14)-mediated cell migration [31], while the latter can inhibit tumor cell growth and migration and, simultaneously, induce the autophagic process of stromal cells, thus exerting a double action that results in a sharp oncosuppressive function [32]. We also reported the downregulation of VASP, an assembly regulatory protein interacting with ZYX, and regulating cell adhesion and migration [33]. Interestingly, the absence of VASP/ZYX interaction impairs cell adhesion and promotes cell migration and invasion. Moreover, VASP affects the phosphorylation of focal adhesion kinase (FAK), which has anti-apoptotic properties, thus being involved in the regulation of the apoptotic process [33]. Tumor-derived exosomes carry on their surface specific tumor markers. Disialoganglioside GD2 is a marker associated with neuroectoderm-derived cancers and has been identified on the surface of exosomes isolated from melanoma [54] and neuroblastoma patients [8]. Among the Exo-prot cargo, we were not able to observe specific enzymes involved in the ganglioside metabolic pathway differentially expressed between the NB and CTRL subjects. This result was also confirmed by pathway analysis, which did not reveal the enrichment of ganglioside synthesis/catabolism processes.

The comparison between HR-NB and LR-NB patients identified Exo-prots upregulated and downregulated in patients with HR-NB disease. Network and pathway analyses revealed that such proteins are connected and act in common biological systems, exerting a major function in pathways that are not only involved in oncogenesis, such as inflammation, cell proliferation and immune response regulation, but also in the acquisition of tumor aggressive traits, such as loss of cell adhesion, migration and angiogenesis. In particular, we observed the upregulation of MYH9, which regulates cell polarity and cytoskeleton [36,37] and has been shown to enhance cell migration in breast cancer cells [37]. Indeed, the exosome-mediated transport of MYH9 is able to directly promote macrophage migration, resulting in increased macrophage infiltration at the tumor site and, consequently, in cancer cell metastasis formation [36,37]. The significantly deregulated Exo-prots in the HR-NB patients included molecules involved in ECM interaction, a key aspect for the establishment of cell-to-cell communication, with a major impact on the definition of the tumor microenvironment required for tumor growth and dissemination. Interestingly, among the upregulated Exo-prots in the HR-NB patients, we identified FN1 and LTBP1. The former is a glycoprotein that exerts its oncogenic role mainly by upregulating cell proliferation and inhibiting apoptosis [38,55] and by regulating angiogenesis, contributing to the direction of blood vessel formation and stimulating cell differentiation toward an endothelial phenotype [38]. LTBP1 is an extracellular matrix protein that enhances cell migration and invasion and promotes the epithelial-to-mesenchymal transition (EMT), a process at the basis of the acquisition of cell motility and invasion properties, leading to metastasis formation [39]. In esophageal squamous cell carcinoma, a positive correlation between FN1 and LTBP1 has been demonstrated [39]. Indeed, while LTBP1 was mainly expressed in tumor tissues, FN1 was highly expressed in fibroblasts of the stroma, suggesting a potential correlation between the overexpression of the two genes, which synergistically act to induce EMT. Among the downregulated proteins, we identified CALR, a protein localized in the endoplasmic reticulum and surveilling proper protein folding, cell adhesion, integrin signaling and antigen presentation [40]. CALR exposed on the surface of tumor cells can trigger phagocytosis and initiate the anticancer immune response [40]. Hence, the downregulation of CALR can have oncogenic effects because of the impairment of physiological homeostasis and the reduced stimulation of immune surveillance against cancer cells [40]. We also observed downregulation of AKAP12, a scaffold protein belonging to the A-kinase anchoring protein (AKAP) family and regulating signal transduction cascade. In particular, AKAP members drive the compartmentalization of cAMP-dependent protein kinases (PKAs) to different subcellular locations to ensure the specificity of PKA-mediated signal transduction [56]. AKAP12 exerts a tumor-suppressive function through the negative modulation of angiogenesis, cell proliferation and migration by preventing PKC activation [57]. Thus, reduced expression of AKAP12 may result in the activation of pro-tumoral and metastatic pathways.

Importantly, ROC analysis demonstrated that the exosomal levels of NCAM, NCL, LUM and VASP have significant diagnostic power in discriminating NB patients from CTRL subjects. High levels of NCAM and NCL have already been described as unfavorable prognostic factors in NB [53,58], and herein we reported, for the first time, their significant upregulation within the exosomes derived from plasma samples of NB patients at diagnosis. In addition, our results indicated that MYH9, FN1, CALR, AKAP12 and, with a lower performance, LTBP1 expression can discriminate HR-NB and LR-NB patients with high specificity and sensitivity. Interestingly, FN1 overexpression has been associated with mesenchymal phenotype [59], which confers chemoresistance to NB cells. Thus, the exosome-mediated delivery of such proteins that we reported to be overexpressed in HR-NB patients could be involved in therapy-resistant mechanisms. Our results showed that the combination of all Exo-prot diagnostic and prognostic markers outperformed each single marker. Therefore, measuring the whole panel of such molecules will provide higher specificity and sensitivity to the analysis. Importantly, the overexpression of NCL and NCAM and the downregulation of DCN and LUM in the NB patient-derived exosomes compared to the CTRL samples was confirmed also in primary tumor tissues. Consistency between the proteomics data and tumor gene expression was also reported for the upregulation of FN1 and the downregulation of CALR and AKAP12 in the HR-NB patients compared to the LR-NB cases. The lack of correspondence between exosomes and tumor tissues for VASP, LGALS3BP, MYH9 and LTBP1 could be due to the fact that these vesicles are loaded through active mechanisms [60]. Thus, the differential expression specifically observed in the exosomes may be due to the cargo selection, which does not necessarily reflect the tissue gene expression levels. Functional experiments and a prospective clinical trial should be performed to increase our knowledge of the biological mechanisms involved in NB formation and development and to robustly assess the applicability of Exo-prots for improving NB patient diagnosis and prognosis.

## 5. Conclusions

We characterized the Exo-prot cargo of HR, LR-NB and CTRL subjects and identified significantly differentially expressed molecules. We assessed the diagnostic significance of specific deregulated Exo-prots and we identified highly accurate markers differentiating NB vs. CTRL samples and HR vs. LR-NB patients. The present study is the first, to the best of our knowledge, to characterize the protein content of exosomes derived from the plasma samples of NB patients. The results showed that these vesicles contain differentially expressed Exo-prots that are mainly involved in cancer-associated pathways and in the acquisition of aggressive tumor features, thus playing a major role in driving oncogenesis and tumor progression. Importantly, we demonstrated that a few specific Exo-prots are able to discriminate, with high accuracy, NB patients from CTRL samples, whereas others are able to precisely distinguish HR-NB from LR-NB patients. These results provide evidence of the potential relevance of liquid biopsies to integrate the diagnosis of NB tumors and improving risk stratification to help refine patient treatment.

## Figures and Tables

**Figure 1 cells-12-02516-f001:**
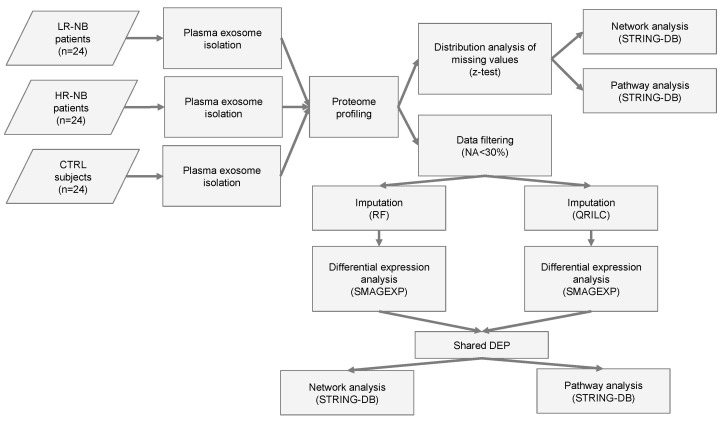
Flowchart of the analysis. Flow chart of the NB (n = 48) and CTR (n = 24) plasma sample analysis. NB included LR-NB (n = 24) and HR-NB (n = 24) patients. The diagram reports the different steps of the bioinformatic analysis applied to the proteomic data, which includes the distribution analysis of missing data, data filtering based on the number of missing values, the application of two different imputation methods (RF and QRILC), differential expression analyses, the selection of the Exo-prots commonly differentially expressed between the RF and QRILC datasets and pathway and network analyses.

**Figure 2 cells-12-02516-f002:**
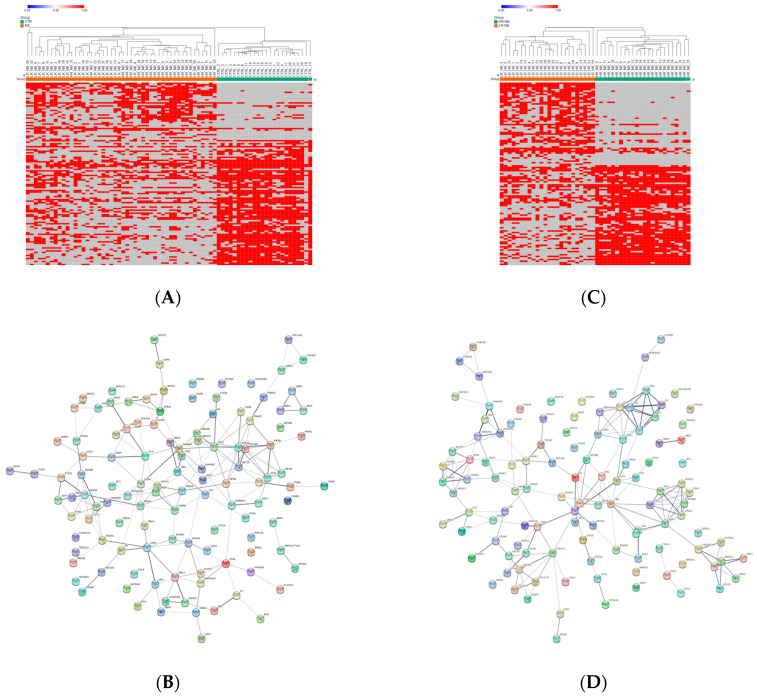
Distribution analysis of the detectable and missing expression values across samples. (**A**) Heatmap showing the significantly differentially represented Exo-prots (n = 121) between the NB and CTRL samples according to the Z-test statistics based on the distribution of missing values. Exo-prots with Z-test values larger than 2.6 or lower than −2.6 were considered to have a significantly different distribution of missing values between groups. The heatmap was drawn using global coloring. Detectable values are colored in red and missing values in grey. A dendrogram is reported above the plot. (**B**) Protein–protein interaction network among the 121 differentially represented Exo-prots. A network was built using the STRING-DB tool. The line thickness indicates the strength of data support. A medium confidence of 0.4 was set up as the minimum required interaction score. (**C**) Heatmap showing the significantly differentially represented Exo-prots (n = 115) between the HR-NB and LR-NB samples according to the Z-test statistics. Exo-prots with Z-test values larger than 2.6 or lower than −2.6 were considered to have a significantly different distribution of missing values across groups. The heatmap was drawn using global coloring. Detectable values are colored in red and missing values in grey. A dendrogram is reported above the plot. (**D**) Protein–protein interaction network among the 115 differentially represented Exo-prots. A network was built using String-DB. The line thickness indicates the strength of the data support. A medium confidence of 0.4 was set up as the minimum required interaction score.

**Figure 3 cells-12-02516-f003:**
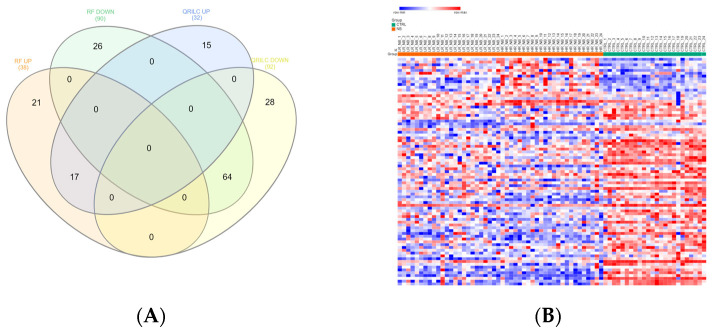
Significantly differentially expressed Exo-prots between the NB and CTRL samples. (**A**) Venn diagram showing the number of commonly and exclusively up- and downregulated Exo-prots in the RF and QRILC datasets. (**B**,**C**) Heatmaps showing the expression values of the deregulated Exo-prots identified in the RF (panel **B**) and QRILC (panel **C**) datasets. The NB and CTRL group labels are shown above the plot. A color key is reported in the top left part of the plot. (**D**) Protein–protein interaction network showing the interactions among the 81 differentially represented Exo-prots between the NB and CTRL samples and shared by the RF and QRILC datasets. A network was built using the STRINGg-DB tool. The Exo-prots are shown as nodes. The line width indicates the strength of data support. A medium confidence of 0.4 was set up as the minimum required interaction score.

**Figure 4 cells-12-02516-f004:**
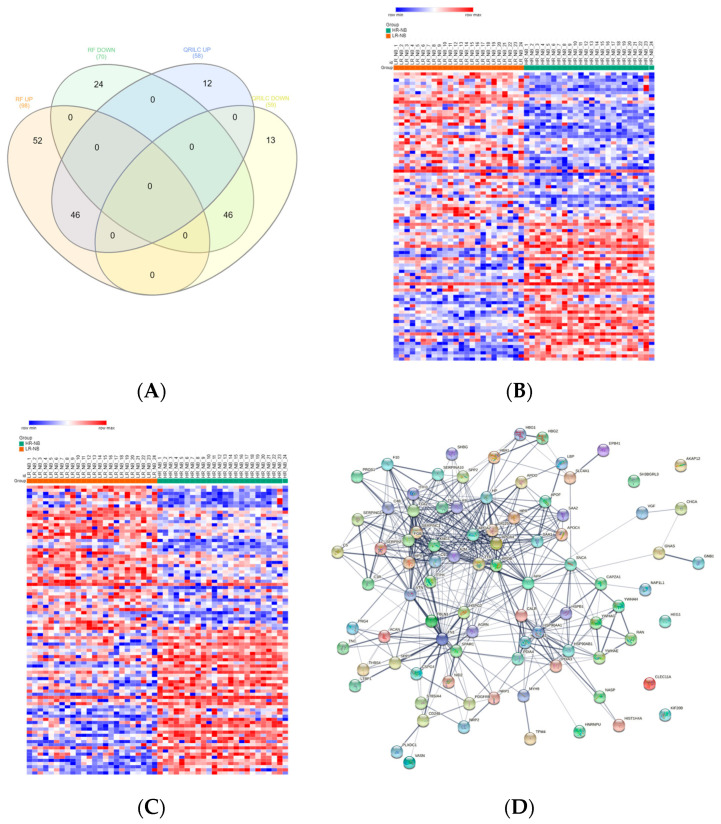
Significantly differentially expressed Exo-prots between the HR-NB and LR-NB samples. (**A**) Venn diagram showing the number of commonly and exclusively up- and downregulated Exo-prots in the RF and QRILC datasets by comparing the HR-NB and LR-NB samples. (**B**,**C**) Heatmaps showing the expression values of the deregulated Exo-prots identified in the RF (panel **B**) or QRILC (panel **C**) dataset. The HR-NB and LR-NB group labels are shown above the plot. A color key is reported in the top left part of the plot. (**D**) Protein–protein interaction network showing the interactions among the 92 differentially represented Exo-prots. Network analysis showing the interactions among the 92 commonly differentially expressed proteins between the HR-NB and LR-NB samples and shared by the RF and QRILC datasets. The network was built using the STRING-DB tool. Exo-prots are shown as nodes. The line width indicates the strength of data support. A medium confidence of 0.4 was set up as the minimum required interaction score.

**Figure 5 cells-12-02516-f005:**
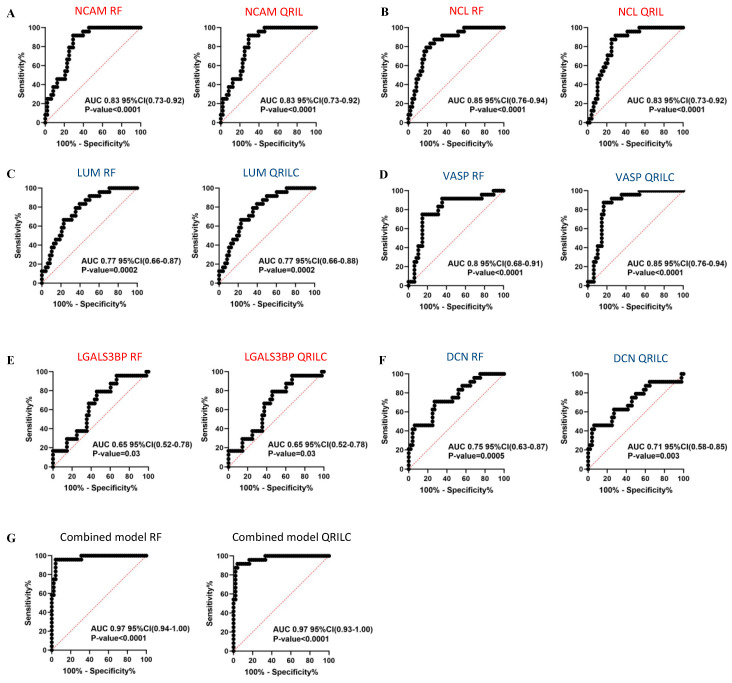
Results of the ROC analysis to assess the diagnostic value of the selected differentially expressed Exo-prots between the NB and CTRL subjects. ROC analysis was performed to assess the diagnostic value of the differentially expressed Exo-prots between the NB and CTRL samples. Analysis was carried out using the expression values from the RF or QRILC datasets. ROC curves for *NCAM* (**A**), *NCL* (**B**), *LUM* (**C**), *VASP* (**D**), *LGALS3BP* (**E**) and *DCN* (**F**) singularly and the combination of all markers (**G**) are reported. Sensitivity (%) is shown on the *Y*-axis, 100%; specificity (%) is shown on the *X*-axis. Areas under the curves (AUCs), coincidence intervals (CIs) and *p*-values are reported for each graph. The red dotted line corresponds to AUC = 0.5, representing a random classifier. *p*-Values lower than 0.05 are considered statistically significant. Red and blue labels refer to the upregulated and downregulated Exo-prots in the NB vs. CTRL samples, respectively. The gene symbol and dataset name are reported above each plot.

**Figure 6 cells-12-02516-f006:**
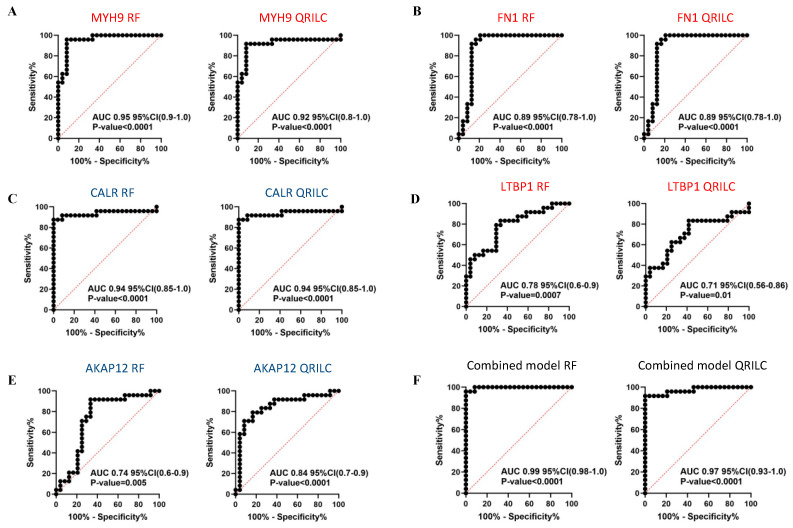
Results of the ROC analysis to assess the prognostic value of the selected differentially expressed Exo-prots between the HR-NB and LR-NB samples. ROC analysis was performed to assess the prognostic value of selected Exo-prot in discriminating between the HR-NB and LR-NB patients. Analysis was carried out using the expression values from the RF and QRILC datasets. ROC curves for MYH9 (**A**), FN1 (**B**), CALR (**C**), LTBP1 (**D**) and AKAP12 (**E**) singularly and the combination of all markers (**F**) are reported. Sensitivity (%) is shown on the *Y*-axis, 100%; specificity (%) is shown on the *X*-axis. Areas under the curves (AUCs), coincidence intervals (CIs) and *p*-values are reported for each graph. The red dotted line corresponds to AUC = 0.5, representing a random classifier. *p*-Values lower than 0.05 are considered statistically significant. Red and blue labels refer to upregulated and downregulated Exo-prots in the HR-NB vs. LR-NB patients, respectively. The gene symbol and dataset name are reported above each plot.

**Figure 7 cells-12-02516-f007:**
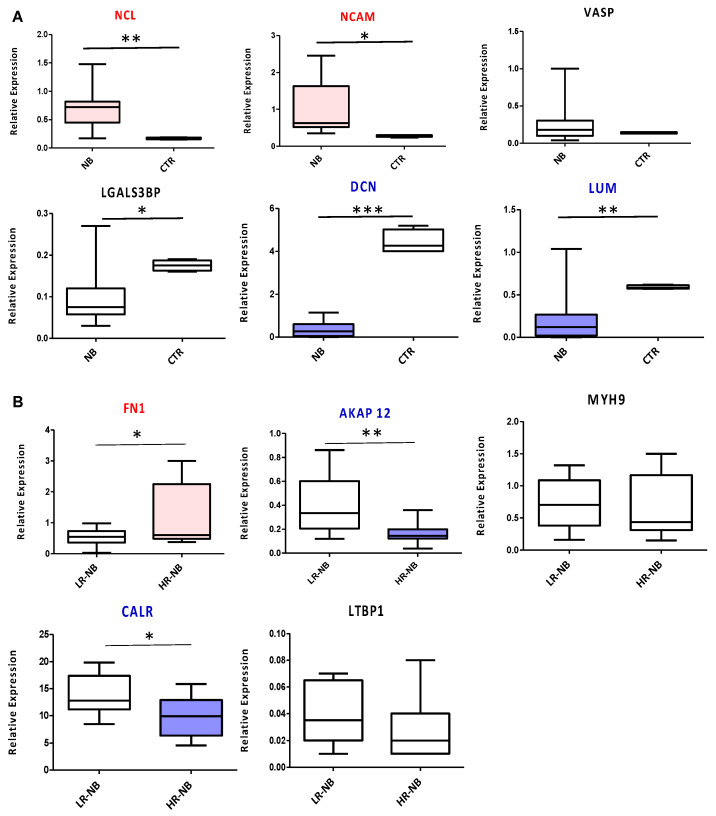
RTqPCR validation of the genes encoding the diagnostic or prognostic Exo-prots identified by ROC analysis. Transcript levels were compared between the HR-NB (n = 10), LR-NB (n = 10) tissues and CTRL subjects (**A**) and between the HR-NB and LR-NB patients (**B**). The graph shows the mean of 2^(−DCT)^ values, against the mean of the DCt of the reference gene *GAPDH*. The results are shown as a box plot and are expressed as the mean normalized gene expression values. The boxes show the values that fall between the 25th and 75th percentiles, the horizontal lines represent the mean values, and the whiskers (lines that extend from the boxes) represent the highest and lowest values for each group. *p*-Values of NB relative to CTRL: * *p* < 0.05; ** *p* < 0.01; *** *p* < 0.001. *p*-Values of HR-NB relative to LR-NB: * *p* < 0.05; ** *p* < 0.01.

**Table 1 cells-12-02516-t001:** Clinical features of the NB and CTRL subjects included in this study.

	Cohort (*n* = 72)			
	Patients (*n* = 48)		Controls (*n* = 24)	
	n	%	n	%
**Sex**				
Male	28	59	16	66
Female	20	41	8	34
**Age at diagnosis**				
<18 months	19	40	7	30
≥18 months	29	60	17	70
**INSS * stage**				
1	9	19	-	-
2	7	15	-	-
4	8	16	-	-
4S	24	50	-	-
**MYCN status**				
Amplified	18	37	-	-
Not amplified	26	54	-	-
N/A	5	9	-	-
**Relapse**				
Yes	7	15	-	-
No	41	85	-	-
**Event overall**				
Yes	34	71	-	-
No	14	29	-	-

* INSS: International Neuroblastoma Staging System.

**Table 2 cells-12-02516-t002:** Selection of the significant pathways of deregulated Exo-prots in the NB patients according to analysis of the distribution of missing values.

Database ^a^	Pathway ^b^	Gene Count ^c^	FDR ^d^
GO BP	Immune system process	42	6.50 × 10^−7^
GO BP	Immune effector process	24	6.68 × 10^−6^
GO BP	Neutrophil degranulation	15	2.10 × 10^−4^
GO BP	Myeloid leukocyte activation	16	2.70 × 10^−4^
GO BP	Response to stimulus	75	3.20 × 10^−4^
GO BP	Cell activation involved in immune response	16	5.40 × 10^−4^
GO BP	Activation of immune response	12	1.30 × 10^−3^
GO BP	Actin filament organization	10	1.30 × 10^−3^
GO BP	Regulation of cell–substrate adhesion	9	2.00 × 10^−3^
GO BP	Leukocyte activation	18	3.30 × 10^−3^
GO BP	Actin filament-based process	14	3.80 × 10^−3^
GO BP	Actin cytoskeleton organization	13	4.00 × 10^−3^
GO BP	Positive regulation of integrin-mediated signaling pathway	3	5.20 × 10^−3^
GO BP	Complement activation	5	5.70 × 10^−3^
GO BP	Cytoskeleton organization	19	9.60 × 10^−3^
GO BP	Complement activation, lectin pathway	3	1.04 × 10^−2^
GO BP	Regulation of focal adhesion assembly	5	1.04 × 10^−2^
GO BP	Regulation of cell migration	16	1.27 × 10^−2^
GO BP	Regulation of actin filament polymerization	7	1.70 × 10^−2^
GO BP	Cell migration	16	1.71 × 10^−2^
GO BP	Complement activation, classical pathway	4	1.73 × 10^−2^
GO BP	Regulation of cell adhesion	14	1.73 × 10^−2^
GO BP	Positive regulation of cell–substrate adhesion	6	1.86 × 10^−2^
GO BP	Regulation of cytoskeleton organization	12	1.86 × 10^−2^
GO BP	Cell motility	17	1.97 × 10^−2^
GO BP	Positive regulation of neuron migration	3	2.32 × 10^−2^
GO BP	Extracellular matrix organization	9	2.89 × 10^−2^
GO BP	Regulation of cell junction assembly	7	2.89 × 10^−2^
GO BP	Inflammatory response	11	3.40 × 10^−2^
GO BP	Regulation of substrate adhesion-dependent cell spreading	4	3.78 × 10^−2^
GO BP	Actin filament bundle assembly	4	4.15 × 10^−2^
GO BP	Positive regulation of cell adhesion molecule production	2	4.77 × 10^−2^
GO BP	Positive regulation of extracellular exosome assembly	2	4.77 × 10^−2^
KEGG	Focal adhesion	9	1.50 × 10^−3^
KEGG	ECM–receptor interaction	6	2.50 × 10^−3^
Reactome	Immune system	35	6.19 × 10^−6^
Reactome	Innate immune system	23	3.97 × 10^−5^
Reactome	Cell–extracellular matrix interaction	4	2.20 × 10^−3^
Reactome	Lectin pathway of complement activation	3	5.30 × 10^−3^
Reactome	Complement cascade	5	6.90 × 10^−3^

GO BP, KEGG and Reactome pathway analyses were carried out with STRING-DB on Exo-prots exclusively expressed in the NB or CTRL samples. The pathways are listed by increasing false discovery rate values for each ontology. ^a^ Name of the ontology defining a pathway. GO BP stands for Gene Ontology biological process. KEGG stands for Kyoto Encyclopedia of Genes and Genomes. ^b^ Official name of a GO biological process, KEGG or Reactome pathway. ^c^ Number of identified genes. ^d^ FDR (false discovery rate) shows *p*-values corrected for multiple testing within each category using the Benjamini–Hochberg procedure. Values lower than 0.05 are considered significant.

**Table 3 cells-12-02516-t003:** Relative expression of the commonly regulated Exo-Prots between the NB and CTRL samples in the RF and QRILC datasets.

Protein Name	Gene Name ^a^	RF_logFC ^b^	RF_adj *p*-Value ^c^	QRILC_logFC ^d^	QRILC_adj *p*-Value ^e^
**Upregulated**					
Nucleolin	NCL	3.30	9.80 × 10^−6^	3.46	3.90 × 10^−5^
Cathepsin G	CTSG	2.51	1.20 × 10^−3^	2.66	4.90 × 10^−3^
C-reactive protein	CRP	2.31	4.80 × 10^−5^	2.06	4.80 × 10^−2^
Tubulin beta-1 chain	TUBB1	2.28	8.10 × 10^−4^	2.56	8.60 × 10^−4^
Serum amyloid A-1 protein	SAA1	2.17	4.20 × 10^−3^	2.69	1.80 × 10^−2^
Histone H4	HIST1H4A	1.96	3.00 × 10^−2^	1.96	3.10 × 10^−2^
Heterogeneous nuclear ribonucleoproteins C1/C2	HNRNPC	1.60	2.00 × 10^−3^	2.83	7.00 × 10^−5^
Prothymosin alpha	PTMA	1.41	3.70 × 10^−3^	2.64	7.70 × 10^−5^
Nuclear autoantigenic sperm protein	NASP	1.31	4.10 × 10^−2^	2.03	7.20 × 10^−3^
Heterogeneous nuclear ribonucleoprotein U	HNRNPU	1.20	1.40 × 10^−2^	1.91	4.60 × 10^−3^
Neural cell adhesion molecule 1	NCAM1	0.92	1.20 × 10^−4^	0.92	2.80 × 10^−4^
Complement C4-B	C4B	0.81	2.00 × 10^−3^	0.81	3.50 × 10^−3^
Golgi membrane protein 1	GOLM1	0.81	9.20 × 10^−3^	1.04	3.50 × 10^−3^
Platelet glycoprotein Ib alpha chain	GP1BA	0.77	3.50 × 10^−4^	0.77	8.40 × 10^−4^
Alpha-2-macroglobulin	A2M	0.75	3.00 × 10^−2^	0.75	3.40 × 10^−2^
Plasma protease C1 inhibitor	SERPING1	0.62	4.10 × 10^−2^	0.62	4.70 × 10^−2^
Galectin-3-binding protein	LGALS3BP	0.59	2.50 × 10^−2^	0.59	3.10 × 10^−2^
**Downregulated**					
Alpha-1B-glycoprotein	A1BG	−0.77	6.40 × 10^−4^	−0.61	3.20 × 10^−2^
Endosialin	CD248	−0.60	3.60 × 10^−2^	−1.44	1.20 × 10^−2^
Lumican	LUM	−0.65	1.20 × 10^−3^	−0.65	2.80 × 10^−3^
78 kDa glucose-regulated protein	HSPA5	−0.66	3.50 × 10^−3^	−0.76	1.20 × 10^−2^
Haptoglobin-related protein	HPR	−0.66	3.00 × 10^−2^	−2.09	4.30 × 10^−2^
Protein disulfide–isomerase A3	PDIA3_DR2	−0.68	1.30 × 10^−3^	−0.72	3.50 × 10^−2^
Aggrecan core protein	ACAN	−0.73	1.10 × 10^−2^	−0.95	2.90 × 10^−2^
Ig gamma-1 chain C region	IGHG1	−0.73	3.10 × 10^−2^	−0.86	2.20 × 10^−2^
Complement factor B	CFB	−0.75	4.80 × 10^−2^	−0.87	4.70 × 10^−2^
Dentin sialophosphoprotein	DSPP	−0.76	4.00 × 10^−3^	−1.73	2.10 × 10^−4^
Flavin reductase (NADPH)	BLVRB	−0.84	4.50 × 10^−2^	−1.42	2.00 × 10^−2^
Coronin	CORO1A	−0.86	1.20 × 10^−3^	−1.78	2.40 × 10^−3^
Apolipoprotein A-II	APOA2	−0.86	4.20 × 10^−5^	−0.86	1.40 × 10^−4^
Alpha-1-antichymotrypsin	SERPINA3	−0.89	4.00 × 10^−3^	−0.89	5.80 × 10^−3^
Complement component C8 beta chain	C8B	−0.89	3.30 × 10^−3^	−0.89	4.80 × 10^−3^
Decorin	DCN	−0.93	7.20 × 10^−4^	−0.92	4.30 × 10^−2^
Vasodilator-stimulated phosphoprotein	VASP	−1.00	1.20 × 10^−3^	−2.80	1.90 × 10^−6^
Protein 4.1	EPB41	−1.01	1.20 × 10^−2^	−3.16	3.50 × 10^−7^
Plexin-B1	PLXNB1	−1.01	4.30 × 10^−4^	−0.91	3.90 × 10^−2^
Protein Z-dependent protease inhibitor	SERPINA10	−1.07	1.10 × 10^−2^	−2.44	3.90 × 10^−3^
Vasorin	VASN	−1.10	8.60 × 10^−5^	−2.20	6.00 × 10^−5^
Transforming growth factor-beta-induced protein ig-h3	TGFBI	−1.11	1.80 × 10^−3^	−1.56	9.20 × 10^−3^
GTP-binding nuclear protein Ran	RAN	−1.13	9.50 × 10^−3^	−1.86	4.60 × 10^−3^
Angiopoietin-related protein 6	ANGPTL6	−1.14	3.30 × 10^−4^	−1.44	5.70 × 10^−4^
Clusterin	CLU	−1.15	5.80 × 10^−6^	−1.15	1.70 × 10^−5^
Clathrin light chain A	CLTA	−1.16	4.00 × 10^−3^	−1.21	2.90 × 10^−2^
Latent-transforming growth factor-beta-binding protein 1	LTBP1	−1.19	1.90 × 10^−3^	−1.96	3.50 × 10^−4^
Eukaryotic translation initiation factor 5	EIF5	−1.22	5.70 × 10^−4^	−2.19	8.90 × 10^−5^
Alpha-synuclein	SNCA	−1.28	9.10 × 10^−4^	−1.67	9.90 × 10^−3^
Matrix Gla protein	MGP	−1.28	9.10 × 10^−4^	−1.17	1.20 × 10^−2^
Interleukin-7 receptor subunit alpha	IL7R	−1.33	3.60 × 10^−8^	−1.81	3.00 × 10^−4^
Melanocyte protein PMEL	PMEL	−1.34	5.40 × 10^−5^	−1.21	1.20 × 10^−2^
Plexin domain-containing protein 2	PLXDC2	−1.35	1.70 × 10^−9^	−2.02	4.70 × 10^−6^
Osteomodulin	OMD	−1.36	1.00 × 10^−7^	−1.06	4.90 × 10^−3^
Band 3 anion transport protein	SLC4A1	−1.38	2.40 × 10^−2^	−1.69	1.10 × 10^−2^
Vitamin K-dependent protein S	PROS1	−1.39	4.20 × 10^−5^	−1.39	7.70 × 10^−5^
Collectin-11	COLEC11	−1.41	2.00 × 10^−5^	−1.47	2.80 × 10^−3^
Transitional endoplasmic reticulum ATPase	VCP	−1.44	1.20 × 10^−4^	−1.44	2.10 × 10^−4^
CD44 antigen	CD44	−1.46	4.20 × 10^−5^	−1.23	4.00 × 10^−3^
14-3-3 protein beta/alpha	YWHAB	−1.46	2.70 × 10^−4^	−1.49	4.00 × 10^−3^
Hemoglobin subunit beta	HBB	−1.48	2.50 × 10^−3^	−1.48	3.40 × 10^−3^
Chondroitin sulfate proteoglycan 4	CSPG4	−1.48	1.40 × 10^−4^	−1.23	8.00 × 10^−3^
Hemoglobin subunit alpha	HBA1	−1.55	8.10 × 10^−3^	−1.55	8.80 × 10^−3^
Alpha-2-HS-glycoprotein	AHSG	−1.55	2.80 × 10^−6^	−1.55	6.60 × 10^−6^
UBE2O	UBE2O	−1.56	2.10 × 10^−5^	−2.15	2.50 × 10^−4^
Asporin	ASPN	−1.70	1.20 × 10^−7^	−2.33	6.60 × 10^−6^
Peptidase inhibitor 16	PI16	−1.70	3.40 × 10^−4^	−1.88	6.80 × 10^−4^
Thrombospondin-4	THBS4	−1.74	2.70 × 10^−5^	−1.74	4.90 × 10^−5^
Importin subunit beta-1	KPNB1	−1.81	4.70 × 10^−7^	−1.82	9.40 × 10^−6^
Heparin cofactor 2	SERPIND1	−1.89	1.80 × 10^−5^	−1.89	2.80 × 10^−5^
Transcription initiation factor TFIID subunit 9	TAF9	−1.93	2.30 × 10^−9^	−1.69	4.30 × 10^−2^
Uncharacterized protein C14orf37	C14orf37	−1.95	1.60 × 10^−7^	−1.33	1.50 × 10^−2^
Complement C1s subcomponent	C1S	−2.14	4.50 × 10^−10^	−2.14	1.70 × 10^−9^
Hemoglobin subunit delta	HBD	−2.51	1.90 × 10^−5^	−3.96	5.50 × 10^−6^
Phosphatidylinositol-glycan-specific phospholipase D	GPLD1	−2.55	3.20 × 10^−18^	−4.65	1.70 × 10^−12^
Thrombospondin-3	THBS3	−2.61	1.30 × 10^−8^	−2.61	6.70 × 10^−6^
Lipopolysaccharide-binding protein	LBP	−2.69	5.80 × 10^−6^	−3.27	2.30 × 10^−5^
Serum paraoxonase/arylesterase 1	PON1	−3.18	2.90 × 10^−7^	−3.87	2.70 × 10^−6^
Spectrin beta chain	SPTB	−3.47	2.20 × 10^−7^	−4.18	3.00 × 10^−6^
Secreted phosphoprotein 24	SPP2	−3.76	2.10 × 10^−10^	−3.46	3.50 × 10^−7^
Sex hormone-binding globulin	SHBG	−4.26	8.50 × 10^−16^	−4.64	3.40 × 10^−14^
Complement C1r subcomponent	C1R	−5.48	9.90 × 10^−12^	−5.84	1.50 × 10^−11^
Spectrin alpha chain	SPTA1	−5.77	7.80 × 10^−11^	−6.54	3.20 × 10^−11^
Ankyrin-1	ANK1	−6.35	2.20 × 10^−10^	−6.56	3.30 × 10^−10^

Exo-prot expression profile was evaluated in the NB and CTRL samples, and comparative analysis of the expression differences between the two groups was carried out by a two-sample Student’s *t*-test after imputation by the RF or QRILC methods. Exo-prots are listed by decreasing FC value. ^a^ Name of the protein-coding genes. ^b^ Fold change values expressed as log2 after the RF imputation method. FC values greater than 0.67 are reported. ^c^ *p*-Value adjusted for FDR (false discovery rate). Adjusted *p*-values lower than 0.05 are considered significant. ^d^ Fold change values expressed as log2 after the RF imputation method. FC values greater than 0.67 are reported. ^e^ *p*-Value adjusted for FDR (false discovery rate). Adjusted *p*-values lower than 0.05 are considered significant.

**Table 4 cells-12-02516-t004:** Selection of significant pathways of the Exo-prots regulated in the comparison between the NB and CTRL samples.

Database ^a^	Pathway ^b^	Gene Count ^c^	FDR ^d^
GO Process	Complement activation	9	4.93 × 10^−8^
GO Process	Regulation of complement activation	9	4.93 × 10^−8^
GO Process	Leukocyte-mediated immunity	16	6.90 × 10^−6^
GO Process	Immune effector process	19	9.04 × 10^−6^
GO Process	Immune system process	30	1.35 × 10^−5^
GO Process	Acute inflammatory response	7	2.37 × 10^−5^
GO Process	Immune response	23	3.36 × 10^−5^
GO Process	Regulation of immune effector process	12	7.75 × 10^−5^
GO Process	Regulation of immune system process	21	1.90 × 10^−4^
GO Process	Regulation of immune response	16	2.10 × 10^−4^
GO Process	Acute-phase response	5	4.40 × 10^−4^
GO Process	Activation of immune response	10	9.70 × 10^−4^
GO Process	Inflammatory response	11	1.70 × 10^−3^
GO Process	Myeloid leukocyte activation	11	4.40 × 10^−3^
GO Process	Negative regulation of intrinsic apoptotic signaling pathway in response to DNA damage	3	3.83 × 10^−2^
GO Process	Regulation of cell death	17	4.27 × 10^−2^
KEGG	ECM–receptor interaction	5	5.30 × 10^−3^
Reactome	Extracellular matrix organization	10	8.84 × 10^−5^
Reactome	ECM proteoglycans	6	1.40 × 10^−4^
Reactome	Degradation of the extracellular matrix	5	1.90 × 10^−2^
Reactome	Neutrophil degranulation	8	3.73 × 10^−2^

GO BP, KEGG and Reactome pathway analyses were carried out with STRING-DB on the Exo-prots differentially expressed in the NB vs. CTRL samples. Pathways are listed by increasing false discovery rate values for each ontology. ^a^ Name of the ontology defining a pathway. GO BP stands for Gene Ontology biological process. KEGG stands for Kyoto Encyclopedia of Genes and Genomes. ^b^ Official name of a GO biological process, KEGG or Reactome pathway. ^c^ Number of identified genes. ^d^ FDR (false discovery rate) shows *p*-values corrected for multiple testing within each category using the Benjamini–Hochberg procedure. Values lower than 0.05 are considered significant.

**Table 5 cells-12-02516-t005:** Selection of significant pathways of the Exo-prots exclusively detectable in the HR-NB or LR-NB patients.

Database ^a^	Pathway ^b^	Gene Count ^c^	FDR ^d^
GO BP	Immune system process	47	2.98 × 10^−10^
GO BP	Response to stress	55	5.40 × 10^−10^
GO BP	Immune response	34	2.55 × 10^−8^
GO BP	Activation of immune response	16	9.11 × 10^−7^
GO BP	Innate immune response	18	3.97 × 10^−5^
GO BP	Inflammatory response	15	9.06 × 10^−5^
GO BP	Acute inflammatory response	7	9.95 × 10^−5^
GO BP	Regulation of actin filament-based process	13	1.20 × 10^−4^
GO BP	Lymphocyte-mediated immunity	8	3.80 × 10^−4^
GO BP	Regulation of focal adhesion assembly	6	4.30 × 10^−4^
GO BP	Cell migration	17	2.20 × 10^−3^
GO BP	Positive chemotaxis	5	2.40 × 10^−3^
GO BP	Actin filament organization	9	2.50 × 10^−3^
GO BP	Regulation of actin filament organization	9	3.80 × 10^−3^
GO BP	Actin cytoskeleton organization	12	5.00 × 10^−3^
GO BP	Regulation of cell adhesion	14	6.60 × 10^−3^
GO BP	Cytoskeleton organization	18	7.70 × 10^−3^
GO BP	Chemotaxis	11	2.58 × 10^−2^
GO BP	Leukocyte migration	8	3.81 × 10^−2^
GO BP	Reactive oxygen species metabolic process	5	3.84 × 10^−2^
KEGG	ECM–receptor interaction	6	1.90 × 10^−3^
Reactome	Complement cascade	7	4.39 × 10^−5^
Reactome	Regulation of complement cascade	6	1.80 × 10^−4^
Reactome	Metabolism of proteins	28	1.20 × 10^−3^
Reactome	ECM proteoglycans	5	9.40 × 10^−3^
Reactome	Apoptosis	6	2.72 × 10^−2^

GO, KEGG and Reactome pathway analyses were carried out with STRING-DB on Exo-prots exclusively expressed in the HR-NB or LR-NB samples. Pathways are listed by increasing false discovery rate values for each ontology. ^a^ Name of the ontology defining a pathway. GO BP stands for Gene Ontology biological process. KEGG stands for Kyoto Encyclopedia of Genes and Genomes. ^b^ Official name of a GO biological process, KEGG or Reactome pathway. ^c^ Number of identified genes. ^d^ FDR (false discovery rate) shows *p*-values corrected for multiple testing within each category using the Benjamini–Hochberg procedure. Values lower than 0.05 are considered significant.

**Table 6 cells-12-02516-t006:** Relative expression of the commonly regulated Exo-prots between the HR-NB and LR-NB patients in the RF and QRILC datasets.

Protein Name	Gene Name ^a^	RF_logFC ^b^	RF_adj *p*-Value ^c^	QRILC_logFC ^d^	QRILC_adj *p*-Value ^e^
**Upregulated**					
Myosin-9	MYH9	4.34	2.20 × 10^−9^	3.96	8.90 × 10^−6^
Complement C1r subcomponent	C1R	3.42	5.40 × 10^−6^	2.70	3.30 × 10^−3^
Hemoglobin subunit gamma-1	HBG1	2.95	9.10 × 10^−5^	4.61	3.90 × 10^−5^
Hemoglobin subunit gamma-2	HBG2	2.78	3.30 × 10^−4^	3.24	4.40 × 10^−3^
Hemoglobin subunit alpha	HBA1	2.67	4.90 × 10^−5^	2.67	8.50 × 10^−5^
Band 3 anion transport protein	SLC4A1	2.43	7.50 × 10^−6^	1.76	9.60 × 10^−3^
Neuroendocrine secretory protein 55	GNAS	2.43	5.00 × 10^−5^	2.58	5.70 × 10^−4^
Secreted phosphoprotein 24	SPP2	2.33	2.30 × 10^−4^	2.33	4.10 × 10^−4^
Nidogen-2	NID2	2.20	8.00 × 10^−5^	1.94	2.10 × 10^−2^
Fibronectin	FN1	2.17	2.40 × 10^−6^	2.17	7.10 × 10^−6^
Heat shock protein beta-1	HSPB1	2.10	2.60 × 10^−5^	2.95	6.10 × 10^−5^
Fibulin-1	FBLN1	2.10	2.10 × 10^−6^	1.96	2.60 × 10^−3^
SPARC	SPARC	1.79	1.40 × 10^−5^	2.27	8.10 × 10^−5^
Apolipoprotein D	APOD	1.68	7.90 × 10^−9^	1.27	4.20 × 10^−3^
Vasorin	VASN	1.63	5.40 × 10^−10^	2.26	1.60 × 10^−4^
F-actin-capping protein subunit alpha-1	CAPZA1	1.57	8.00 × 10^−5^	1.73	3.30 × 10^−3^
Antithrombin-III	SERPINC1	1.54	5.70 × 10^−6^	2.28	8.90 × 10^−6^
Apolipoprotein A-IV	APOA4	1.51	1.10 × 10^−4^	1.51	2.60 × 10^−4^
C-type lectin domain family 11 member A	CLEC11A	1.51	2.40 × 10^−6^	1.51	8.90 × 10^−6^
Basement membrane-specific heparan sulfate proteoglycan core protein	HSPG2	1.48	8.90 × 10^−4^	1.48	1.50 × 10^−3^
Hemopexin	HPX	1.45	1.90 × 10^−7^	3.35	1.70 × 10^−11^
Latent-transforming growth factor-beta-binding protein 1	LTBP1	1.32	8.10 × 10^−4^	1.35	3.80 × 10^−2^
GTP-binding nuclear protein Ran	RAN	1.31	4.00 × 10^−3^	1.95	1.40 × 10^−2^
Neuropilin-1	NRP1	1.29	2.00 × 10^−6^	1.09	4.40 × 10^−3^
Serotransferrin	TF	1.27	5.30 × 10^−3^	2.06	3.20 × 10^−3^
SH3 domain-binding glutamic acid-rich-like protein 3	SH3BGRL3	1.25	2.80 × 10^−2^	2.31	8.40 × 10^−3^
Chondroitin sulfate proteoglycan 4	CSPG4	1.23	3.90 × 10^−3^	1.23	5.60 × 10^−3^
Platelet-derived growth factor receptor beta	PDGFRB	1.22	1.10 × 10^−4^	1.20	9.10 × 10^−3^
Osteopontin	SPP1	1.21	1.50 × 10^−5^	1.21	6.00 × 10^−5^
Neurosecretory protein VGF	VGF	1.21	7.20 × 10^−5^	1.22	1.20 × 10^−2^
Guanine nucleotide-binding protein G(I)/G(S)/G(T) subunit beta-1	GNB1	1.19	6.60 × 10^−3^	2.16	1.10 × 10^−2^
Thrombospondin-4	THBS4	1.18	1.10 × 10^−2^	1.18	1.40 × 10^−2^
Alpha-2-macroglobulin	A2M	1.09	5.30 × 10^−3^	1.09	8.00 × 10^−3^
Protein HEG homolog 1	HEG1	1.04	2.40 × 10^−4^	1.04	6.80 × 10^−4^
Aggrecan core protein	ACAN	1.03	1.30 × 10^−3^	1.36	5.20 × 10^−3^
Alpha-synuclein	SNCA	1.00	2.50 × 10^−2^	1.75	2.30 × 10^−2^
Sex hormone-binding globulin	SHBG	0.95	3.20 × 10^−2^	1.33	8.60 × 10^−3^
Tenascin	TNC	0.92	3.20 × 10^−2^	0.92	4.40 × 10^−2^
Amyloid beta A4 protein	APP	0.91	1.50 × 10^−2^	0.91	2.00 × 10^−2^
Plexin domain-containing protein 1	PLXDC1	0.86	3.60 × 10^−5^	1.78	1.40 × 10^−5^
Protein 4.1	EPB41	0.85	3.70 × 10^−2^	1.76	8.00 × 10^−3^
Agrin	AGRN	0.80	1.40 × 10^−2^	0.80	2.00 × 10^−2^
Vitronectin	VTN	0.80	6.50 × 10^−5^	0.80	3.80 × 10^−4^
Kininogen-1	KNG1	0.73	9.80 × 10^−4^	1.32	7.90 × 10^−4^
Transthyretin	TTR	0.68	1.70 × 10^−2^	0.68	2.70 × 10^−2^
Endosialin	CD248	0.62	3.80 × 10^−2^	2.12	1.50 × 10^−3^
**Downregulated**					
CMP-N-acetylneuraminate-poly-alpha-2,8-sialyltransferase	ST8SIA4	−0.59	1.80 × 10^−2^	−1.07	4.90 × 10^−3^
Apolipoprotein A-II	APOA2	−0.62	3.40 × 10^−3^	−0.62	8.20 × 10^−3^
Protein disulfide-isomerase A3	PDIA3	−0.74	7.10 × 10^−3^	−0.74	1.20 × 10^−2^
Vitamin K-dependent protein S	PROS1	−0.79	3.40 × 10^−2^	−0.79	4.90 × 10^−2^
Haptoglobin-related protein	HPR	−0.88	5.30 × 10^−3^	−5.65	3.50 × 10^−9^
Endoplasmin	HSP90B1	−0.89	2.10 × 10^−4^	−0.89	7.20 × 10^−4^
Complement C4-B	C4B	−0.90	1.10 × 10^−3^	−0.90	2.70 × 10^−3^
Coagulation factor X	F10	−1.15	8.40 × 10^−6^	−1.15	4.00 × 10^−5^
14-3-3 protein eta	YWHAH	−1.16	3.90 × 10^−2^	−1.87	1.80 × 10^−2^
Fibrinogen beta chain	FGB	−1.17	3.30 × 10^−4^	−1.17	7.70 × 10^−4^
Neuropilin-2	NRP2	−1.25	8.20 × 10^−4^	−1.25	1.50 × 10^−3^
Inter-alpha-trypsin inhibitor heavy chain H3	ITIH3	−1.26	1.40 × 10^−10^	−1.26	5.00 × 10^−9^
Ig kappa chain C region	IGKC	−1.26	6.30 × 10^−5^	−1.06	1.30 × 10^−2^
Coagulation factor IX	F9	−1.36	2.10 × 10^−3^	−1.36	3.30 × 10^−3^
Inter-alpha-trypsin inhibitor heavy chain H4	ITIH4	−1.40	6.90 × 10^−3^	−1.40	9.10 × 10^−3^
Proteoglycan 4	PRG4	−1.41	3.90 × 10^−4^	−2.67	1.50 × 10^−5^
Plasma protease C1 inhibitor	SERPING1	−1.42	1.10 × 10^−7^	−1.42	8.10 × 10^−7^
Protein Z-dependent protease inhibitor	SERPINA10	−1.47	2.30 × 10^−4^	−3.60	2.50 × 10^−4^
14-3-3 protein epsilon	YWHAE	−1.49	1.90 × 10^−3^	−1.49	3.10 × 10^−3^
Alpha-2-antiplasmin	SERPINF2	−1.50	9.50 × 10^−3^	−1.95	1.70 × 10^−2^
A-kinase anchor protein 12	AKAP12	−1.51	9.30 × 10^−3^	−2.79	2.50 × 10^−4^
Fibrinogen gamma chain	FGG	−1.57	3.60 × 10^−3^	−1.57	4.90 × 10^−3^
Nucleosome assembly protein 1-like 1	NAP1L1	−1.70	4.80 × 10^−3^	−1.79	4.90 × 10^−2^
Ig alpha-1 chain C region	IGHA1	−1.76	2.50 × 10^−4^	−1.76	5.00 × 10^−4^
Alpha-1-antichymotrypsin	SERPINA3	−1.85	1.30 × 10^−9^	−1.85	8.30 × 10^−9^
Nuclear autoantigenic sperm protein	NASP	−1.86	1.60 × 10^−2^	−2.27	1.00 × 10^−2^
14-3-3 protein gamma	YWHAG	−1.89	1.40 × 10^−5^	−1.89	3.40 × 10^−5^
Apolipoprotein B-100	APOB	−1.93	8.00 × 10^−6^	−1.93	2.00 × 10^−5^
Apolipoprotein F	APOF	−2.08	2.80 × 10^−7^	−2.43	4.70 × 10^−3^
Complement C5	C5	−2.23	3.80 × 10^−7^	−5.20	4.30 × 10^−7^
Protein disulfide-isomerase A4	PDIA4	−2.25	1.20 × 10^−9^	−2.25	4.90 × 10^−9^
Calreticulin	CALR	−2.29	7.50 × 10^−7^	−2.39	1.70 × 10^−6^
Apolipoprotein C-IV	APOC4	−2.39	5.00 × 10^−10^	−3.94	7.00 × 10^−10^
Heterogeneous nuclear ribonucleoprotein U	HNRNPU	−2.41	3.80 × 10^−6^	−3.26	6.10 × 10^−7^
Ig mu chain C region	IGHM	−2.44	5.60 × 10^−10^	−2.44	2.50 × 10^−9^
Serum amyloid A-2 protein	SAA2	−2.56	1.40 × 10^−5^	−4.37	1.30 × 10^−4^
Chromogranin-A	CHGA	−2.75	1.60 × 10^−5^	−2.75	3.40 × 10^−5^
Haptoglobin	HP	−2.79	6.10 × 10^−4^	−2.79	9.30 × 10^−4^
Heat shock protein HSP 90-beta	HSP90AB1	−3.03	1.80 × 10^−4^	−3.03	3.00 × 10^−4^
Tropomyosin alpha-4 chain	TPM4	−3.05	3.90× 10^−4^	−3.05	6.50 × 10^−4^
Serum amyloid A-1 protein	SAA1	−3.23	9.90 × 10^−5^	−4.83	8.50 × 10^−5^
Complement component C9	C9	−3.34	2.10 × 10^−10^	−5.32	9.60 × 10^−13^
Lipopolysaccharide-binding protein	LBP	−3.68	6.50 × 10^−12^	−4.85	2.70 × 10^−11^
Heat shock protein HSP 90-alpha	HSP90AA1	−3.92	2.10 × 10^−10^	−3.92	4.70 × 10^−10^
Kinesin-like protein KIF20B	KIF20B	−3.96	1.30 × 10^−9^	−3.31	1.30 × 10^−3^
Histone H4	HIST1H4A	−4.75	7.50 × 10^−7^	−4.75	1.50 × 10^−6^

Exo-prot expression profiles were evaluated in the HR-NB and LR-NB samples, and comparative analysis of the expression differences between the two sample groups was carried out by a two-sample Student’s *t*-test after imputation by the RF or QRILC method. Exo-prots are listed by decreasing FC value. ^a^ Name of the protein-coding genes. ^b^ Fold change values expressed as log2 after the RF imputation method. FC values greater than 0.67 are reported. ^c^ *p*-Value adjusted for FDR (false discovery rate). Adjusted *p*-values lower than 0.05 are considered significant. ^d^ Fold change values expressed as log2 after the QRILC imputation method. FC values greater than 0.67 are reported. ^e^ *p*-Value adjusted for FDR (false discovery rate). Adjusted *p*-values lower than 0.05 are considered significant.

**Table 7 cells-12-02516-t007:** Selection of significant pathways of the Exo-prots regulated in the comparison between the HR-NB and LR-NB samples.

Database ^a^	Pathway ^b^	Gene Count ^c^	FDR ^d^
GO BP Process	Negative regulation of endopeptidase activity	17	2.81 × 10^−12^
GO BP Process	Acute inflammatory response	11	1.00 × 10^−10^
GO BP Process	Regulation of peptidase activity	19	1.57 × 10^−10^
GO BP Process	Extracellular matrix organization	16	1.94 × 10^−9^
GO BP Process	Inflammatory response	18	8.16 × 10^−9^
GO BP Process	Acute-phase response	8	2.45 × 10^−8^
GO BP Process	Regulation of complement activation	8	1.04 × 10^−7^
GO BP Process	Positive regulation of cell motility	16	8.80 × 10^−7^
GO BP Process	Positive regulation of cell migration	15	2.95 × 10^−6^
GO BP Process	Immune system process	32	3.34 × 10^−6^
GO BP Process	Movement of cell or subcellular component	23	2.54 × 10^−5^
GO BP Process	Regulation of cell motility	18	2.55 × 10^−5^
GO BP Process	Regulation of cell migration	17	4.52 × 10^−5^
GO BP Process	Positive regulation of cell communication	25	4.70 × 10^−5^
GO BP Process	Blood vessel morphogenesis	12	5.03 × 10^−5^
GO BP Process	Blood vessel development	13	5.81 × 10^−5^
GO BP Process	Biological adhesion	17	1.10 × 10^−4^
GO BP Process	Complement activation, classical pathway	5	1.40 × 10^−4^
GO BP Process	Regulation of cell death	23	1.60 × 10^−4^
GO BP Process	Regulation of transforming growth factor-beta production	5	1.60 × 10^−4^
GO BP Process	Positive regulation of cell–substrate adhesion	7	2.00 × 10^−4^
GO BP Process	Response to cytokines	18	2.00 × 10^−4^
GO BP Process	Locomotion	19	2.80 × 10^−4^
GO BP Process	Vascular endothelial growth factor signaling pathway	4	3.20 × 10^−4^
GO BP Process	Cell adhesion	16	3.60 × 10^−4^
GO BP Process	Positive regulation of chemotaxis	7	5.00 × 10^−4^
GO BP Process	Regulation of substrate adhesion-dependent cell spreading	5	5.00 × 10^−4^
GO BP Process	Cell activation	17	5.30 × 10^−4^
GO BP Process	Regulation of cell–substrate adhesion	8	5.30 × 10^−4^
GO BP Process	Neural crest cell migration involved in autonomic nervous system development	3	5.40 × 10^−4^
GO BP Process	Angiogenesis	9	1.00 × 10^−3^
GO BP Process	Positive regulation of substrate adhesion-dependent cell spreading	4	2.20 × 10^−3^
GO BP Process	Regulation of cell adhesion	12	5.00 × 10^−3^
GO BP Process	Toll-like receptor signaling pathway	5	5.70 × 10^−3^
GO BP Process	Positive regulation of endothelial cell migration	5	6.20 × 10^−3^
GO BP Process	Ventral trunk neural crest cell migration	2	1.17 × 10^−2^
GO BP Process	Telomerase holoenzyme complex assembly	2	1.17 × 10^−2^
GO BP Process	Cell–matrix adhesion	5	1.23 × 10^−2^
GO BP Process	Vascular endothelial growth factor receptor signaling pathway	4	1.24 × 10^−2^
GO BP Process	Regulation of leukocyte migration	6	1.55 × 10^−2^
GO BP Process	Negative regulation of cell death	13	2.03 × 10^−2^
GO BP Process	Actin cytoskeleton organization	9	2.04 × 10^−2^
KEGG	PI3K-Akt signaling pathway	13	1.56 × 10^−6^
KEGG	ECM–receptor interaction	7	2.75 × 10^−5^
KEGG	Focal adhesion	6	1.85 × 10^−2^
Reactome	Extracellular matrix organization	16	2.12 × 10^−10^
Reactome	Innate immune system	22	2.01 × 10^−7^
Reactome	ECM proteoglycans	8	7.05 × 10^−7^
Reactome	Degradation of the extracellular matrix	5	1.43 × 10^−2^

GO BP, KEGG and Reactome pathway analyses were carried out with STRING-DB on the Exo-prots differentially expressed in the HR-NB and LR-NB samples. Pathways are listed by increasing false discovery rate values for each ontology. ^a^ Name of the ontology defining a pathway. GO BP stands for Gene Ontology biological process. KEGG stands for Kyoto Encyclopedia of Genes and Genomes. ^b^ Official name of a GO biological process, KEGG or Reactome pathway. ^c^ Number of identified genes. ^d^ FDR (false discovery rate) shows *p*-values corrected for multiple testing within each category using the Benjamini–Hochberg procedure. Values lower than 0.05 are considered significant.

## Data Availability

The proteomic data have been deposited to the ProteomeXchange Consortium via the PRIDE [61] partner repository with the dataset identifier PXD042422.

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
