# Peer review of "Plasma-Derived Exosome Proteins as Novel Diagnostic and Prognostic Biomarkers in Neuroblastoma Patients"

_cells, 2023, doi:10.3390/cells12212516_

Round 1
Reviewer 1 Report
In this manuscript, the authors explore the potential biomarkers and key biological associated with tumor development in NB patients via performed proteomic analysis of exosomes isolated from plasma samples of NB patients and age-matched CTRL sample. The authors discovered a different Exo-prot expression profile between NB and CTRL patients, with the modulation of proteins involved in cell migration, proliferation and metastasis. The authors present lots of bioinformatic analysis, however, lacking the most important experiments to confirm the results. I think the author should at least present some experiments to confirm the results. Such as:
1) The authors should try to test whether the NCAM, NCL, LUM and VASPA have a diagnostic value in discriminating NB patients from CTRL via using NB cell lines and normal cell lines.
2) The authors should try to test whether the MYH9, FN1, AKAP12 and LTBP1 can differentiate HR-NB and LR-NB patients via using the relative NB cell lines.
In general, I believe that these concerns should be addressed prior to publication in Cells.
Author Response
"Please see the attachment"

Reviewer 2 Report
1. In method section “Exosomes isolation and proteomics analysis”.
Its not clearly mentioned, how much plasma were used from three different group for the exosome isolation. Then only, it could be compared among three different groups.
Also, how did author confirm, whether, it is exosome or something else. It is very important to confirm it before proceeding with other experiments. Better techniques for confirmation is electron microscopy and western blot using exosome markers.
Usually, exosomes are in 40-15nm in size, and microvesicles also ranging from 30-1000nm, If it is not confirmed it as exosomes, then its very hard to say whether it is exosomes or microvesicles.
2. In introduction section, I would suggest to please define the exosomes about size and how it is synthesized and secreted in plasma. For this, I would suggest to cite the recent publication such as Karn et al., 2021 (https://www.mdpi.com/2227-9059/9/10/1373) and Zhang et al., 2021 (https://link.springer.com/article/10.1186/s12882-021-02417-8), in the introduction section especially before “ Exosomes are defined as the bioprint of the cell of origin and, thus, the study of their content allows to gain accurate information of primary tumor features [6–8].
Overall, the work is very interesting.
its fine
Author Response
"Please see the attachment."

Reviewer 3 Report
I believe that this paper is valuable in examining the diagnosis and prognosis of Neuroblastoma through Exosome analysis. Before acceptance, I would be happy if you could consider two points.
First, the ROC analysis is conducted only with a single marker, but I speculate that the accuracy might improve if it is conducted in a composite manner (for example, with NCAM and NCL in Figure 5). Second, in the control samples, there is an outlier sample (CTRL18), and I thought it would be good to explain its significance. Please consider these points.
Minor editing of English language required.
Author Response
"Please see the attachment."

Reviewer 4 Report
Interesting paper that shows data for the expression of different proteins in exosomes isolated from people with LR or HR neuroblastoma as well as an age- and sex-matched control population.
Just a few points to address or consider:
1. There should be some discussion of GD2 since it is an NB marker albeit not necessarily for risk but for tumor presence and relapse. It has also been shown to be present in exosomes isolated from patients with melanoma (Yesmin et al doi: 10.1038/s41598-023-31216-4). Recognizing that you were looking at proteins, were any of those involved in ganglioside synthesis/catabolism (such as B3GalT4 found in sEVs) differentially expressed in exosomes from either HR- or LR-NB patients vs controls?
2. How was purity of exosome preparations ascertained?
3. In footnotes to tables (eg. 2, 5) showing values for FDRs, the statement is made that they are expressed as -log10 however it looks like they are just given in scientific notation. Please clarify.
4. Line 420: what is ttyy?
5. In Fig.5 and 6 define what the red lines represent.
Author Response
"Please see the attachment."

Round 2
Reviewer 1 Report
Accept
Reviewer 2 Report
The authors addressed all my comments. Therefore, I recommend it for publication.